# Hydroxypropyl-Beta Cyclodextrin Barrier Prevents Respiratory Viral Infections: A Preclinical Study

**DOI:** 10.3390/ijms25042061

**Published:** 2024-02-08

**Authors:** Angela Lu, Brandon Ebright, Aditya Naik, Hui L. Tan, Noam A. Cohen, Jean-Marie C. Bouteiller, Gianluca Lazzi, Stan G. Louie, Mark S. Humayun, Isaac Asante

**Affiliations:** 1Alfred E. Mann School of Pharmacy and Pharmaceutical Sciences, University of Southern California, Los Angeles, CA 90089, USA; alu52904@usc.edu (A.L.); bebright@usc.edu (B.E.); aanaik@usc.edu (A.N.); slouie@usc.edu (S.G.L.); 2Department of Otorhinolaryngology, Head and Neck Surgery, University of Pennsylvania, Philadelphia, PA 19104, USA; lihtan@pennmedicine.upenn.edu (H.L.T.); ncohen@pennmedicine.upenn.edu (N.A.C.); 3Corporal Michael J. Crescenz VA Medical Center, Philadelphia, PA 19104, USA; 4Viterbi School of Engineering, University of Southern California, Los Angeles, CA 90007, USA; jbouteil@usc.edu (J.-M.C.B.); lazzi@usc.edu (G.L.); humayun@med.usc.edu (M.S.H.); 5Department of Ophthalmology, Keck School of Medicine, University of Southern California, Los Angeles, CA 90089, USA

**Keywords:** COVID-19, SARS-CoV-2, cyclodextrin, viral replication, inflammation

## Abstract

The emergence and mutation of pathogenic viruses have been occurring at an unprecedented rate in recent decades. The coronavirus disease 2019 (COVID-19) pandemic caused by severe acute respiratory syndrome coronavirus 2 (SARS-CoV-2) has developed into a global public health crisis due to extensive viral transmission. In situ RNA mapping has revealed angiotensin-converting enzyme 2 (ACE2) expression to be highest in the nose and lower in the lung, pointing to nasal susceptibility as a predominant route for infection and the cause of subsequent pulmonary effects. By blocking viral attachment and entry at the nasal airway using a cyclodextrin-based formulation, a preventative therapy can be developed to reduce viral infection at the site of entry. Here, we assess the safety and antiviral efficacy of cyclodextrin-based formulations. From these studies, hydroxypropyl beta-cyclodextrin (HPBCD) and hydroxypropyl gamma-cyclodextrin (HPGCD) were then further evaluated for antiviral effects using SARS-CoV-2 pseudotypes. Efficacy findings were confirmed with SARS-CoV-2 Delta variant infection of Calu-3 cells and using a K18-hACE2 murine model. Intranasal pre-treatment with HPBCD-based formulations reduced viral load and inflammatory signaling in the lung. In vitro efficacy studies were further conducted using lentiviruses, murine hepatitis virus (MHV), and influenza A virus subtype H1N1. These findings suggest HPBCD may be used as an agnostic barrier against transmissible pathogens, including but not limited to SARS-CoV-2.

## 1. Introduction

Respiratory viral infections, highly prevalent in recent decades, burden the United States healthcare system with a direct cost estimate of over USD 16 billion [1,2]. Epidemic and pandemic outbreaks like the severe acute respiratory syndrome coronavirus (SARS-CoV), H1N1, and the Middle East Respiratory Syndrome Coronavirus (MERS-CoV) have had worldwide impacts in 2002, 2009, and 2012, respectively [3,4,5]. Recently, seasonal infections of SARS-CoV-2, respiratory syncytial virus (RSV), and influenza viruses further illustrate the impact of viral infections on the economy. The rapid evolution of variants with high transmissibility and pathogenic virulence has compounded the persistence of these infections. In all these respiratory viral infections, the primary site of infection is the mucocutaneous membrane. This is the first line of defense against aerosol-transmissible pathogens, which when compromised, increases host susceptibility to clinical disease. To reduce SARS-CoV-2 infection, it is critical to develop effective therapeutic strategies to reduce viral transmission.

Although SARS-CoV-2 infection has clinically been linked to downstream respiratory symptoms, the nasal lining serves as the primary gateway for viral entry into the human host [6]. Cellular entry of SARS-CoV-2 is mediated by viral spike (S) protein receptor binding domain (RBD) interaction with angiotensin-converting enzyme 2 (ACE2), a carboxypeptidase found on the surface of epithelial cells [7,8]. In situ RNA mapping has revealed high ACE2 expression in the nasal airway, further supporting the nasal epithelium as the predominant initial site for SARS-CoV-2 infection [9]. Furthermore, scRNA-seq datasets reveal ACE2 and transmembrane protease serine 2 (TMPRSS2) to be highly expressed in the ciliated and goblet cells of the nasal cavity [6]. Following infection of ciliated epithelial cells, viral replication and luminal release into the respiratory secretions allow for advancement and spread into the respiratory tract to advance pulmonary disease [10]. Since the nasal passage serves as the initial site of infection from which the virus spreads, the development of a preventative nasal spray intervention capable of enhancing the mucocutaneous barrier may be an effective strategy to inhibit viral infection.

The ability to enhance mucocutaneous defense is an alternative strategy to agnostically prevent pathogenic attachment, which is required for viral infection. Cyclodextrins (CDs) are sugar polymers structurally similar to native mucin glycoproteins and are compatible with nasal epithelial cells [11]. Known to interact and sequester viral membrane cholesterol, cyclodextrins have demonstrated antiviral properties against enveloped viruses by disrupting lipid rafts and cell cholesterol depletion [12,13,14]. This study describes the screening and characterization of the antiviral effects of various types of cyclodextrins to prevent and reduce the severity of mucocutaneous viral infection. In addition, the physiochemical characteristics must meet the features required for intranasal delivery without compromising any physiological function of the target cells. This study contributes to the knowledge of cyclodextrins as antivirals by providing a comparison of different modified beta and gamma-cyclodextrins, as well as by proving the efficacy of HPBCD-based intranasal formulations in a murine SARS-CoV-2 model.

## 2. Results

### 2.1. Cyclodextrin Safety and Efficacy Profiles

Various cyclodextrin analogs, like hydroxypropyl beta-cyclodextrin (HPBCD), hydroxypropyl gamma-cyclodextrin (HPGCD), sulfobutyl ether beta-cyclodextrin (SBECD), and a methyl beta-cyclodextrin (CRYSMEB), were evaluated for their ability to prevent viral attachment and subsequent infection. Table 1 summarizes their physiochemical characteristics concerning their ability to prevent viral attachment and infection. Although cyclodextrins, such as HPBCD and HPGCD, are well tolerated in humans when administered via intravenous and subcutaneous injections, as well as topically, an initial screen was conducted to evaluate any potential cytotoxic effects of these cyclodextrins [15,16]. After 24 h of treatment with 5% and 10% *w*/*v* formulations of the individual CDs, the fluorescent signal of the treated cells for all CD were found to be safe, with cell viability similar to that of the no-treatment (NT) control (Figure 1A). Despite their purported safety, cells treated with 5% *w*/*v* reduced methylated beta-cyclodextrin (CRYSMEB) exhibited cellular morphology changes after treatment, resulting in cell sloughing and detachment. Salem et al. have previously identified an LD50 of 31 mM (~4.5% *w*/*v*) for CRYSMEB in A549 cells, reaffirming this finding [17]. Subsequent safety studies evaluated only HPBCD and HPGCD, where their cellular safety was confirmed using lactate dehydrogenase (LDH) release assay treated for 48 h (Figure 1B). Compared to the no-treatment control, HPBCD and HPGCD did not significantly increase lactate dehydrogenase activity, indicative of cytotoxicity.

The nasal mucosa has a ciliary activity that is important in removing attached particulates entrapped in the mucocutaneous secretions. To assess the impact of these CDs on ciliary activity, primary human nasal epithelial cells cultured using an air–liquid interface (ALI) transwell system were treated with HPBCD and HPGCD [18]. Toxicity was assessed based on ciliary beat frequency (CBF) and percent of ALI culture with an active area of ciliary beating (Figure 1C) [19]. Normal airway cilia beat in a coordinated fashion at an average frequency of 7–16 Hz [20]. Treatment with HPBCD and HPGCD did not alter CBF parameters or significantly reduce the active area across all time points.

### 2.2. Antiviral Effects of Cyclodextrins

To determine the level of viral attachment inhibition, A549-ACE2 cells were treated with increasing concentrations of CDs from 0.5–10% *w*/*v* prior to the addition of the Wuhan D614G mutant SARS-CoV-2 pseudotype. Both HPBCD and HPGCD were able to reduce pseudotype infection by greater than 50% at concentrations as low as 0.5% *w*/*v* (Figure 2A,B). In contrast, SBECD exhibits variable efficacy across increasing concentrations, with 0.5% *w*/*v* significantly reducing intracellular pseudotype infection. Interestingly, higher concentrations > 5% *w*/*v* did not improve SBECD effectiveness (Figure 2C). Here, we show that our CD-based formulations can inhibit earlier variants, such as the Wuhan SARS-CoV-2 variant. One disadvantage of pseudotyped SARS-CoV-2 systems is the lack of replicative potential. As a result, these data show the initial reduction in viral load after SARS-CoV-2 cell penetration but do not take into account viral replication.

SARS-CoV-2 variant emergence over the course of the pandemic has resulted from viral evolutionary properties to evade immune-mediated mechanisms [21,22]. Mutations in the viral spike (S) protein, in particular, amino acid substitutions resulting in an increased electrostatic charge and/or side chain volume, have led to changes in the S receptor binding domain (RBD) surface charge and increased ACE2 binding [23,24,25]. To assess the efficacy of CD-based treatments against viral infection, SARS-CoV-2 pseudotypes were constructed for the Wuhan (D614G mutant), Alpha, Beta, Gamma, Delta, and Epsilon variants using a VSVΔG system [26]. See Appendix A for spike protein characteristics of the SARS-CoV-2 variants. Since viral S protein binding to the cellular ACE2 receptor plays a pivotal role in initiating viral penetration and infection and the pseudotyped virus S protein is similar to that of the native viral S protein, SARS-CoV-2 pseudotypes are widely accepted and used for cellular tropism studies [27].

The infectivity of these SARS-CoV-2 variants was tested using A549-ACE2 with HPBCD or HPGCD from 1–10% *w*/*v*. Both HPBCD and HPGCD were able to significantly reduce the infectivity of the Wuhan variant pseudotype (Figure 3). Interestingly, only HPBCD was able to prevent pseudotype SARS-CoV-2 infection in a concentration-dependent manner across all variants (Figure 3A). Minimal efficacy was observed in HPGCD treatment against the Delta, Cal.20, and P.1 variants. Despite increasing the concentration of HPGCD, it did not improve the ability to inhibit B1.351 and B1.1.7 infections (Figure 3B). In contrast, HPBCD reduced viral infectivity capable of reducing viral infectivity by 92% to 98% across variants at 10% *w*/*v* (Figure 3A). At lower concentrations, such as 5%, HPBCD was able to reduce infection of most variants by 70% or more, except for the B.1.351 and B.1.1.7 variants.

### 2.3. In Silico Modeling of CD Interaction with S Protein

This study shows that not all CDs performed the same. More importantly, the effects observed were dissimilar in terms of safety and antiviral properties. Concentration escalation appears to be important in the ability of CD analogs to prevent viral infection, and only HPBCD demonstrated this type of physicochemical behavior across the various studies. To understand the difference in antiviral effects between HPBCD and HPGCD, in silico molecular docking studies were conducted using AutoDock Vina (version 1.1.2) (Scripps Research Institute, San Diego, CA, USA) to study the interaction of the SARS-CoV-2 S protein with cyclodextrins. The 3D structures of cyclodextrin ligands were generated with ChemDraw Professional and optimized using AutoDock Tools software (version 1.5.6). Spike protein crystal structures were obtained from the Protein Data Bank (PDB), and further optimizations were made in accordance with the AutoDock Vina Manual [28,29]. For each ligand spike protein simulation, the free binding energy (ΔG) of the most favorable conformational output was recorded in Kcal/mol [29]. The dissociation constant (K_d_) was then extrapolated based on the change in free energy (ΔG) measured by the binding score using the following equation:

ΔG = −RTln(K_d_), where R = 1.9872036 cal*K^−1^*mol^−1^ and T = 298 K


The search space region was divided into two segments: (1) the “upper” segment comprising the S1 subunit and the portion of the S2 subunit that comprises the S1/S2 and S2 cleavage sites and (2) the “lower” segment comprising the S2 subunit only. The “upper segment” also contains the RBD and active site cleft.

Across all ligand–protein binding pairs, the binding scores reflective of the free binding energy (ΔG) were similar. AutoDock Vina’s calculated binding score can be interpreted as the sum of all intermolecular interactions present between the ligand and target. Both binding scores and K_d_ were similar for the Wuhan, Alpha, Beta, and Delta variants when assessing S1/S2 subunit binding to HPBCD and HPGCD. The K_d_ of HPGCD for the lower search space region was found to be 6- and 15-fold higher, respectively, for the Alpha and Delta variants when compared to HPBCD, suggesting that HPBCD has greater binding affinity for the spike protein of those variants (Table 2). These observations correspond with more negative binding scores, indicating increased molecular interactions between HPBCD and the S2 subunit of the S proteins. Similarly, The K_d_ of HPGCD for the upper search space region was found to be 20-fold higher in the Epsilon SARS-CoV-2 variant. Moreover, HPGCD docking to Gamma S1/S2 had a 3.3-fold greater binding affinity as compared to HPBCD. This observation is consistent when assessing HPGCD docking to the “lower” S2 subunit alone, where there is a 2.8-fold greater binding affinity for HPGCD compared to HPBCD.

The decreased efficacy of HPGCD compared to HPBCD across variants may be associated with cyclodextrin binding pockets and the molecular binding interactions between the mutated spike proteins and the CDs. Higher K_d_ values for HPGCD-CD interactions in the Alpha, Delta, Epsilon, and Gamma variants provide evidence for this. Additional factors associated with CD efficacy may include cholesterol sequestration capacity or environmental pH. The cholesterol sequestration capacity of HPBCD is at least 75 times more than that of HPGCD [5]. Limitations of the docking data are as follows. Due to the size of the S protein and limitations in search space for the molecular binding study, the protein was separated into two search space regions. The S protein of the Epsilon variant could not be corrected for its missing atoms and as such, the docking scores may be less accurate.

### 2.4. HPBCD and HPGCD Sequester Both Viral and Cellular Cholesterol

To further investigate the potential mechanism of action by which CDs exert their antiviral infectivity effects, A549-ACE2 cells and an rVSV-Luc-SARS-CoV-2-S pseudotype of the Wuhan S protein variant were incubated independently with either HPBCD or HPGCD at 0–10% *w*/*v* formulations. Cells and viral particles were separated from the treatments via a 50 kDa centrifugal filter and quantified using a fluorescence assay kit (Amplex Red Cholesterol Assay) and a cholesterol standard curve. Concentration-dependent decreases in viral-bound cholesterol levels were observed with both HPBCD and HPGCD treatment, and 2.5% *w*/*v* formulations were able to reduce viral cholesterol concentrations by ~3-fold. Over 80% of viral cholesterol is sequestered within 2 h at the 5% *w*/*v* concentration (Figure 4A).

Furthermore, HPBCD and HPGCD were able to reduce membrane-bound cellular cholesterol. A greater than 50% reduction in cell membrane-bound cholesterol is observed with 2.5% HPBCD or HPGCD (*p* < 0.001). While HPBCD also reduces cellular cholesterol in a concentration-dependent manner (Figure 4B), HGBCD had no detectable difference at greater than 2.5% *w*/*v* (Figure 4C). These results may suggest that HPGCD has a greater safety profile compared to HPBCD, as higher concentrations do not deplete cellular cholesterol further.

### 2.5. Cell-Treated and Virus-Treated Cyclodextrin Effect

To confirm whether the CDs had direct interaction with the enveloped viruses, our team assessed the prevention of viral infection by the treatment of cells with a CD-based barrier (cell treated) and the treatment of pseudotyped SARS-CoV-2 with the barrier (virus treated). Cell-treated regimens reflect contraction, whereas virus-treated regimens reflect transmission in the presence and absence of treatment. Pseudotyped viruses treated with CD were added to A549-ACE2 cells to determine the impact of CD treatment on viral transmissibility. rVSV-SARS-CoV-2-Luc (Wuhan variant) were pre-treated with the HPBCD or HPGCD formulations prior to the infection of A549-ACE2 cells. Figure 5A shows the infection of A549-ACE2 under cell-treated conditions, and Figure 5B shows the effect of a 2 h pre-treatment of pseudotyped virus with CDs on the resultant cell infection. Low concentrations of both HPBCD and HPGCD were able to prevent pseudotyped SARS-CoV-2 infection of A549-ACE2 cells dramatically with over 70% inhibition using 0.5% *w*/*v* formulations. Interestingly, the same concentration of both HPBCD and HPGCD (at 0.5% *w*/*v*) incubated with the virus was also able to reduce the infection of A549-ACE2 cells 10-fold and exhibited similar efficacy at higher concentrations. These results suggest that CD-based treatments may prevent viral spread in addition to preventing viral infection of host cells. They also reveal CD interactions with not only the cell but with the SARS-CoV-2 pseudotype. These cell–CD and virus–CD interactions have been confirmed by other authors targeting lipid rafts as a strategy against coronavirus [30,31].

### 2.6. HPBCD Has Broad Effects across Different Enveloped Respiratory Viruses

Consistent with previous SARS-CoV-2 pseudotype study findings, HPBCD was able to reduce lentivirus infection of HEK293T cells in a concentration-dependent manner. These results are similar, irrespective of the viral incubation period (Figure 6A,B). The viral infectivity (percent fluorescence) was normalized to untreated media control for which a 40% reduction was observed in the presence of 2.5% HPBCD. After 2 and 6 h of incubation with the pLV[Exp]-Puro-CMV > EGFP lentivirus, 10% HPBCD is able to reduce infection of HEK293T cells by >50%.

To determine whether HPBCD is an agnostic barrier, wild-type A549 cells were infected with H1N1, an influenza A virus, at an MOI = 0.2 after pre-treatment with either 2.5% HPBCD, 5% HPBCD, or 2.5% HPBCD + 0.5% CMC. Both the 2.5% HPBCD and 5% HPBCD + 0.5% CMC treatment groups were able to significantly reduce intracellular viral load by greater than 90% compared to the no-treatment (NT) control (Figure 6C). These data are promising as they suggest that HPBCD-based treatments reduce both SARS-CoV-2 and influenza infection, two of the major viruses involved in airborne illnesses. A second virus, murine hepatitis virus (MHV) infection, a surrogate beta coronavirus for the study of SARS-CoV-2, of NCTC 1469 cells was compared in the absence and presence of 2.5–10% HPBCD treatment. These studies produced parallel results. Even at the lowest concentration of 2.5% *w*/*v*, a 6–12-fold decrease in viral infectivity was observed compared to the untreated control. At both viral loads, 5000 pfu (MOI = 0.01) and 10,000 pfu (MOI = 0.02), 10% HPBCD is able to inhibit MHV infection fully after 2 h of viral exposure. Similarly, 2.5% *w*/*v* HPBCD is able to reduce infectivity by >50% in both viral load groups (Figure 6D).

These results support our proposal of HPBCD as an agnostic barrier capable of preventing viral attachment and, ultimately, infections. Thus far, efficacy has been observed for lentiviruses, SARS-CoV-2 pseudotypes (across variants), and MHV. This suggests that hydroxypropyl-beta-cyclodextrin is acting as a physical barrier, irrespective of the viral binding receptors found on host cells.

### 2.7. HPBCD Reduces SARS-CoV-2 Infection in Human Bronchial Cells

To corroborate the anti-viral effects observed with CD pre-treatment prior to pseudotyped SARS-CoV-2 infection, confirmatory studies were conducted using SARS-CoV-2 (Delta variant)-infected Calu-3 cells. Calu-3 cells were infected with the SARS-CoV-2 Delta variant for 48 h at an MOI = 0.2 prior to cell lysis with Trizol. Formulations tested include the following: 2.5% HPBCD (*w*/*v*), 5% HPBCD, 10% HPBCD, 5% HPBCD + 0.5% CMC (low viscosity—50–200 cP), 5% HPBCD + 0.5% CMC (high viscosity—1500–3000 cP), 5% HPBCD + 5% HPGCD, and 5% HPBCD + 0.5% CMC (low viscosity + 1 mM zinc sulfate).

Results showed variable SARS-CoV-2 reduction in Calu-3 cells across treatment groups. The 5% HPBCD + 0.5% CMC was able to reduce viral infection by ~50%, whereas 2.5% HPBCD was also able to reduce cellular infection by ~20% (Figure 7A). Both 2.5% HPBCD alone and 5% HPBCD + 0.5% CMC-treated groups had significantly lower viral load (Figure 7B). While 5% HPBCD + 5% HPGCD exhibited similar efficacy compared to the 2.5% HPBCD alone treatment group, previous SARS-CoV-2 pseudotyped data showed variable efficacy of HPGCD across SAR-CoV-2 variants compared to HPBCD (Figure 3). Interestingly, 5% HPBCD alone was not as effective at reducing SARS-CoV-2 infection, suggesting that there may be an upper limit on effective concentration. In preliminary studies with Calu-3 cells, pre-treatment with 0.5% CMC alone showed a 40% reduction, suggesting that increased viscosity of the 5% HPBCD + 0.5% CMC formulation may be driving its effect. Physicochemical changes, such as increasing viscosity, can sustain contact time between the cyclodextrin and the cells, thereby facilitating the sequestration of cholesterol to reduce viral entry [32].

In Vero-E6 cells, ACE2 expression has largely been associated with lipid rafts [33]. Since lipid rafts and cholesterol have been well determined to play a role in SARS-CoV and SARS-CoV-2 entry into cells, we quantified cellular levels of ACE2 in the same study [34,35]. Rather than interfering with ACE2 expression levels, HPBCD preserved ACE2 protein levels moderately, with cell-treated groups having 2- to 3-fold higher levels; 5% HPBCD + 0.5% CMC preserved ACE2 levels significantly (Figure 7C). Moreover, since ACE2 is involved in the renin–angiotensin system (RAS) pathway, gene expression levels of RAS enzymes and receptors were quantified using RTPCR. Interestingly, angiotensin type 2 receptor (AT2R) ΔΔCt levels were decreased with HPBCD treatment, indicating the upregulation of the receptor (AT2R) compared to the untreated control. Interestingly, these results suggest that pre-treatment with HPBCD increases the expression of the AT2R, which combats AT1R pro-inflammatory responses. This study reflects a severe infection scenario with a high MOI and long duration of viral exposure.

### 2.8. HPBCD Reduces SARS-CoV-2 Deposition in Mouse Lung Tissue

To determine whether HPBCD treatment is translatable, an in vivo study was conducted using K18-hACE2 mice (N = 20, male) pre-treated twice (once daily) prior to viral inoculation. Mice were euthanized at 5 dpi and nasal swabs, lung tissue, and blood plasma were collected for end-of-study processing. Throughout the study, the mice maintained their weight with the 5% HPBCD + 0.5% CMC group, experiencing a 5% weight gain compared to the untreated control group. High levels of viral RNA were detected in lung homogenates at 5 dpi, whereas lower levels were found on the nasal swabs as expected. Treatment with the 2.5% HPBCD and 5% HPBCD + 0.5% CMC groups resulted in a 75% and 96% decrease in the viral load found in lung tissue; 5% HPBCD alone was found to be less effective (Figure 8A). These results mimic the in vitro study results conducted using Calu-3 cells, further validating the effects of HPBCD in vivo.

The innate immune system recognizes viral RNA through pattern recognition receptors (PRRs), such as TLR7 [36]. Downstream of the RAS pathway, inflammatory signaling via cytokines and chemokines is mediated through NFkB regulation [37]. The activation of transcriptional factors increases the production of interferons (IFNs), which act as the primary mediators of homeostasis in response to viral infection. All three treatment groups reduced IFNy expression in mouse lung tissue; however, only 5% HPBCD + 0.5% CMC was able to reduce IFNa and IFNb signaling, where IFNb levels were undetectable. Decreases in cytokines and chemokines that regulate virally induced inflammatory responses, such as IL-1b, IL-8, CXCL9, CXCL10, and the CCLs, were observed (Figure 8B). An increase in IL-10 was observed with the 2.5% HPBCD and 5% HPBCD + 0.5% CMC groups.

In addition to reduced gene expression of immune cell regulatory signaling proteins, immune cell recruitment in the lung decreased in the treatment groups, corresponding with the reduced viral load. SARS-CoV-2 infection of K18-hACE2 mice is characterized by progressive and widespread viral pneumonia with alveolar and perivascular inflammation. Histopathological analyses of murine lung tissue show extensive infiltration of immune cells, including neutrophils and mononuclear cell types in the no-treatment group at 5 dpi (Figure 9). A decrease in immune cell count was observed with treatment, as seen in the representative images for each treatment group (not statistically significant). Moreover, compared to mock-infected mice, SARS-CoV-2-infected lung tissue exhibits proteinaceous debris in the alveolar regions, the presence of neutrophils in the interstitial space, and hemorrhages in alveolar spaces [38]. Moreover, the thickening of the alveolar–capillary membrane was observed in the no-treatment and 5% HPBCD groups. The thickening of the perivascular lung tissue in the 2.5% HPBCD and 5% HPBCD + 0.5% CMC-treated mice was significantly reduced, suggesting a less severe disease state.

## 3. Discussion

Viral transmission frequently occurs through direct contact with an infectious individual, respiratory droplets, or environmental aerosols [39,40]. As we continue to battle this highly pathogenic and mutating virus with limited approved treatments, traditional approaches, like vaccines, mask mandates, and social distancing, as well as preventative therapies, remain an untapped opportunity. The dynamic evolution of SARS-CoV-2 and other respiratory viruses has led to the emergence of more transmittable variants that are resistant to current antivirals, immunotherapies, and vaccination strategies [21,41]. Here, we present evidence for the use of HPBCD-based treatments as an effective intervention agnostic to all SARS-CoV-2 variants.

The respiratory mucosa of the nasal passage consists primarily of water and polysaccharides [42]. Cyclodextrins are naturally occurring sugar polymers structurally similar to that of native mucin glycoproteins [43]. Biocompatible with the mucocutaneous lining of the nasal cavity, they interact with membrane and viral lipids to reduce pathogen entry into cells. Some cyclodextrins have been used in parenteral, nasal, and pulmonary product formulations up to 10% (*w*/*v*) with a strong safety profile [44], while others have employed HPBCD as a potential agent to reduce viral infection using in vitro systems, and this study presents their use as a potential therapeutic barrier in both in vitro and in vivo systems [45]. Here, we assess both (1) the ability of HPBCD-based formulations to block initial infection using replication-incompetent pseudotyped SARS-CoV-2 viruses and (2) the efficacy of the barrier in the reduction in viral load in vitro and in vivo using the SARS-CoV-2 Delta variant in a BSL3 setting.

The primary criterion for selecting cyclodextrins was safety, after which efficacy was evaluated. Formulations of CRYSMEB greater than 5% (*w*/*v*) caused cytotoxicity and thus, CRYSMEB formulations were excluded from subsequent SARS-CoV-2 studies. Early in vitro studies using the pseudotyped SARS-CoV-2 Wuhan variant showed that SBECD was unable to reduce the pseudotyped viral infection of cells in a concentration-dependent manner (Figure 2). These effects may be due to the polyanionic R-groups of SBECD, which affect the solubilizing properties. While SARS-CoV-2 spike protein interactions with SBECD have not been studied, the intrinsic activities in the CD complex with small molecules were found to be much lower with SBECD than HPBCD [46]. Sulfo-butyl substitution may hinder SARS-CoV-2 spike protein interactions with the hydrophobic cyclodextrin cavity through structural or ionic interactions. Due to the lack of dose–response relationship observed when SBECD formulations were evaluated using the pseudotyped SARS-CoV-2, further studies were not conducted using these formulations.

While 5% HPBCD alone can reduce initial viral infection by >80%, irrespective of the variant, as shown in our pseudotyped SARS-CoV-2 studies, the 5% HPBCD + 0.5% CMC formulation only reduced viral load by 50% against the SARS-CoV-2 Delta variant at an MOI of 0.2 in Calu3 cells at 48 hpi. This reduction in efficacy observed between a replication-incompetent pseudotyped SARS-CoV-2 versus the SARS-CoV-2 Delta variant is understandable. Viral particles that infect past the barrier continue to replicate, thus reducing the efficacy to a degree. Irrespective of this, these results show that despite a long duration of viral exposure and high MOI, HPBCD is still effective. In a K18-hACE2 knock-in mouse model, 2.5% HPBCD alone and 5% HPBCD + 0.5% CMC significantly reduced viral load in lung tissue by 73% and 94%, respectively. These results suggest that HPBCD-based formulations can lower viral deposition in the lung in a dynamic model, incorporating factors such as intranasal clearance, immune response, and more. In the mice treated with 5% HPBCD + 0.5% CMC, both NFkB and TGFb gene expression is downregulated by 5 dpi. Chemokine and cytokine gene expression levels were also reduced compared to the positive control.

Our studies demonstrated that the HPBCD barrier is safe in clinical and preclinical models. We have previously shown the parameters required to deliver this formulation to cover the intranasal cavity [47]. Short-term exposure up to 48 h of treatment showed no signs of cytotoxicity in our study. No sign of local or systemic toxic effects from nasal administration for 3 months of 10% HPBCD was observed in rabbits or four-day nasal doses of 2.4 g HPBCD in human volunteers [48]. Alboni et al. also evaluated the safety of HPBCD in ACE2 overexpressing human embryonic kidney cells (HEK293T-ACEhi) and observed no apoptotic effect 72 h after overnight exposure to 20 mM HPβCD (equivalent to 3% *w*/*v*). Our studies show that the nasal epithelial cells can withstand higher concentrations of HPBCD than embryonic kidney cells.

Data from this study show viral penetration and safety studies, as well as the impact of cavity size, occupancy, and R-group substitutions on both cholesterol sequestration and cytotoxicity. HPBCD and HPGCD were selected as candidates for antiviral testing. Methyl beta cyclodextrins (MBCDs), such as CRYSMEB, have potential detrimental hemolytic activity; because of these safety concerns, this formulation was eliminated from further considerations [5]. Lactate dehydrogenase (LDH) activity and ciliary beat frequency studies showed that HPBCD and HPGCD have minimal cytotoxicity (Figure 2B). This finding is consistent with beta-cyclodextrin’s (BCD) ability to reduce cell-free cholesterol [49,50], which can deplete lysosomal storage of cholesterol and treat Niemann Pick Disease (NPD) [51,52].

The proposed mechanism of therapeutic action is the interaction of HPBCD with (1) the SARS-CoV-2 S protein and (2) viral membrane cell cholesterol sequestration. CD binding onto the virus itself prevents attachment to hACE2 at the cell surface and subsequent S-ACE2 internalization [35]. Additionally, cellular cholesterol is an important molecular factor involved in the binding of viral surface proteins during viral entry, and viral cholesterol is important for the stability of the viral particles [36]. Here, we see that low concentrations of CD (2.5% *w*/*v*) are able to drastically reduce cholesterol found on the viral membrane by 2- to 3-fold, thereby affecting viral integrity. While SARS-CoV-2 utilizes cholesterol-rich lipid rafts as a platform for viral entry, cellular cholesterol serves a critical function in preserving membrane integrity as well [37]. Raich-Regue et al. have previously identified beta-cyclodextrins from a screening of drug candidates to have antiviral potential, finding methyl beta-cyclodextrin (MBCD) to be effective in a human nasal epithelial model and hamster model up to two days post-infection [53]. Compared to other cyclodextrins, such as 2,6 di-O-methyl beta cyclodextrin (DIMEB) and MBCDs, HPBCD has been shown to have lower cholesterol solubilizing activity at 50 mM [3]. Compared to HPGCD and SBECD, HPBCD exhibited a modestly higher cholesterol solubilizing ability. Overall, these data suggest that compared to MBCDs, HPBCD may exhibit a greater safety profile to facilitate its development into a chronic intermittent use product in humans. To the best of our knowledge, this is the set of data that emphasizes the safety and efficacy of HPBCD in reducing viral deposition in the lung in a hACE2 murine model with up to 5 days of post-infection follow up. Furthermore, we show the added effect of viscosity on 5% HPBCD using CMC.

Limitations and considerations associated with the use of cyclodextrins in a clinical setting exist due to their physiochemical properties. While natural cyclodextrins are poorly soluble in an aqueous solution, chemical substitutions at the 2, 3, and 6 hydroxyl sites greatly increase solubility [54]. While natural beta cyclodextrins only have a solubility of 18.5 mg/mL at 25 °C, HPBCD, a modified CD, has a solubility of >1200 mg/mL [55]. Furthermore, CDs show high stability under alkaline conditions, whereas at a lower pH, CDs may be prone to acid hydrolysis, ring opening, and linearization [56]. However, HPBCD and HPGCD are stable in bases and weak organic acids, but they are concentration- and temperature-dependent strong acid hydrolyses. The storage of the CD formulation in powder for reconstitution form will help to ameliorate stability concerns. Rapid clearance of liquid-based substances from the nasal cavity by ciliated mucosal cells is a significant disadvantage of the intranasal administration of CDs. Modeling approaches are underway to optimize viscosity in order to achieve the most optimal frequency of administration [47].

Despite the need for re-administration, this form of administration is also a less invasive method of delivery that is easily accessible for both pediatric and adult patients. Interspecies testing of tolerable levels of nonclinical vehicles showed that in rats and dogs, intranasal administration of 50 uL/nostril and 1 mL/nostril HPBCD TID 2 h apart for 14 days, respectively, was found to be well tolerated [57]. HPBCD (up to 10% *w*/*v*) has been historically administered in clinical patients six times a day for 4 days [58]. Lastly, based on the proposed mechanism of action of cholesterol sequestration, frequent intranasal dosing may cause cytotoxic effects long term by depleting cholesterol on the mucosal cell surface [59]. Compared to alpha CDs and methylated CDs, modified beta cyclodextrins exhibit reduced cell toxicity [5,60]. Szente et al. identified the cholesterol-solubilizing activity of various cyclodextrin derivatives, whereby HPBCD, HPGCD, and SBECD sequestered significantly less cholesterol; as such, these CDs were originally identified as potential candidates in our studies [5]. High-speed microscopy studies utilizing ciliated nasal epithelial cells and computer simulations are ongoing to understand mucosal clearance of our formulations to provide data in support of a dosing schedule.

In both the in vitro and in vivo studies assessing the efficacy of HPBCD-based formulations against SARS-CoV-2 infection, we see that 2.5% HPBCD and 5% HPBCD + 0.5% CMC reduce viral infection and ameliorate downstream inflammatory responses through the RAS pathway. The efficacy of the 2.5% *w*/*v* treatment group identifies the optimal concentration of HPBCD alone. The efficacy of the 5% HPBCD + 0.5% CMC group is largely driven by the thickening properties of CMC, which may enhance the contact time needed to facilitate cholesterol sequestration. The observed efficacy has not been limited to SARS-CoV-2 but also other enveloped viruses, like influenza and mouse hepatitis virus. This proof of concept can be applied to other seasonal viruses as a strategy to prevent microbial attachment in order to reduce their healthcare burden in the United States.

## 4. Materials and Methods

### 4.1. Production of SARS-CoV-2 Pseudotypes

The following full-length S protein plasmids were kindly provided by Dr. Paula Cannon (USC) along with a VSV-G plasmid for pseudotype production: Wuhan (D614G), Delta, Epsilon (Cal.20), Gamma (P.1), Alpha (B.1.1.7), and Beta (B.1.351). rVSV-ΔG-Luc constructs were made by transfecting pBS-N, pBS-P, pBS-L, and pBS-G plasmids into HEK293T cells. rVSV-ΔG-Luc particles were harvested via ultracentrifugation at 25,000 rpm for 2 h at 4 °C. To produce rVSV-ΔG-Luc bearing the SARS-CoV-2 spike proteins, HEK293T cells were seeded in 10 cm dishes. The recommended cell density is 65–80%. Spike protein plasmids were transfected using CaCl_2_ in 2X HBS. Cell media were changed 6 h post-transfection, followed by rVSV-ΔG-Luc infection. Twenty-four hpi followed the confirmation of cytopathic effects (CPEs), and a supernatant was collected and underwent ultracentrifugation at 25,000 rpm for 2 h at 4 °C using a 20% *w*/*v* sucrose cushion.

### 4.2. Cell Lines

A549-ACE2 lung and HEK293T kidney epithelial cells were maintained in Dulbecco’s modified Eagle medium (DMEM) containing 4.5 g/L D-glucose (Corning, Corning, NY, USA) with 10% fetal bovine serum (FBS; Thermo Fisher Scientific, Waltham, MA, USA) and 1% MEM non-essential amino acids (NEAA; Corning). A549-ACE2 cells were maintained with 2 μg/mL puromycin (InvivoGen, San Diego, CA, USA) as the selection antibiotic. Calu3 cells were maintained in complete Eagle’s Minimum Essential Medium (EMEM; ATCC, Manassas, VA, USA) and supplemented with 10% FBS and 1% NEAA. Cells were incubated at 37 °C with 5% CO_2_. A549-ACE2 and Calu-3 cells were confirmed for ACE2 expression via Western blot prior to experimentation.

### 4.3. Nasal Air–Liquid Interface (ALI) Cultures

Nasal mucosal specimens were obtained via cytologic brushing of patients in the Department of Otorhinolaryngology-Head and Neck Surgery, Division of Rhinology, at the University of Pennsylvania, and the Philadelphia Veteran Affairs Medical Center, after obtaining informed consent. The full study protocol, including the acquisition and use of nasal specimens, was approved by the University of Pennsylvania Institutional Review Board (protocol #800614) and the Philadelphia VA Institutional Review Board (protocol #00781). Patients with a history of systemic disease or on immunosuppressive medications are excluded. ALI cultures were grown and differentiated on 0.4 mm pore transwell inserts as previously described [19,61,62]. In brief, cytologic brush specimens are dissociated and the fibroblast cell population is removed, followed by plating onto transwell inserts. Nasal cells are allowed to grow to confluence in a submerged state (~5 days), and then the apical growth medium is removed. Basal differentiation media is replaced bi-weekly for 3–4 weeks prior to infection. All cultures are subjected to confirmatory tests for differentiation prior to infection; epithelial morphology is monitored via microscopy and ciliation is confirmed. PneumaCult-ALI basal medium (Stemcell Technologies, Vancouver, BC, Canada) was used for differentiation.

### 4.4. VSV Pseudotype Transfection

The human embryonic kidney cell line HEK293T is a well-studied widely used cell line for transfection due to enhanced uptake and protein production efficacy. A variant of the HEK293 cell line, HEK293T cells express the SV40 T antigen ori, greatly enhancing protein expression levels during transfection. HEK293T cells seeded up to 60–70% confluence were transfected in 10 cm dishes with 15 μg of SARS-CoV-2 S plasmid via CaCl_2_ and subsequently infected with VSV-ΔG-Luc virus 24 h later. After 1 h of VSV-ΔG-Luc infection, cells were washed with 1X PBS repeatedly to remove residual VSV-ΔG-Luc. Pseudotype particles were harvested from the supernatant and passed through a 0.45 µm filter after cells exhibited a cytopathic effect (CPE), typically 24–48 h post-infection. The viral supernatant was then ultracentrifuged at 25,000 rpm for 2 h and concentrated 100X in PBS. Relative pseudotype titer (in RLU) was quantified via luciferase assay (Promega, Madison, WI, USA).

### 4.5. Barrier Formulations

The CD-based treatment candidates were formulated and tested using iterative processes to identify safe formulation(s) that can inhibit viral attachment to susceptible cells. Hit formulations were further optimized to widen the therapeutic index by improving cell tolerance while maintaining or improving antiviral attachment. Initial formulations were used to evaluate the safety of the candidates using a concentration escalation ranging from 0 to 20% *w*/*v* of the cyclodextrin formulation. These formulations had a low viscosity consistency to facilitate their utilization as potential candidates for intranasal delivery. The selected formulations were made in distilled water to evaluate pH (ranging from 4.5 to 6.8), thereby guiding their safe use in mammalian cells. Beta (HPBCD, SBECD, and CRYSMEB) and gamma (HPGCD) cyclodextrins were purchased from Roquette (Lestrem, France) and Wacker (Adrian, MI, USA). Formulations with 20% *w*/*v* cyclodextrin were made by dissolving 10 g of each cyclodextrin formulation in 20 mL of distilled water. Solutions were vortexed and made up to 50 mL volume with a complete cell culture medium (DMEM, 10% FBS, 1% NEAA). All formulations underwent 0.22 µm syringe filtration. Cyclodextrin formulations with 0.5% HEC were dissolved in water and heat sterilized due to high viscosity prior to adjustment to the total volume with cell culture medium.

### 4.6. rVSV SARS-CoV-2 Pseudotype Infection Studies

Using a recombinant luciferase-expressing VSV that lacked glycoprotein expression, a D614G-mutated Wuhan spike plasmid was introduced to the virus via transfection to mimic SARS-CoV-2-mediated infection. A549-ACE2 cells were seeded to 100% confluency and pre-incubated with cyclodextrin treatments for 1 h. The treatment was then removed, and cells were washed with PBS prior to replacement with complete cell culture media (DMEM, 10% FBS, 1% NEAA). Cells were infected with rVSV-SARS-CoV-2-Luc pseudotype (100,000 RLU) for 6 h prior to washout. After 24 h of infection, cells were lysed using 1X cell culture lysis buffer. Luciferase assays were conducted as described by the manufacturer (Promega) to detect pseudotype infection of A549-ACE2 cells. Studies across SARS-CoV-2 variants followed the same protocol.

### 4.7. Cholesterol Assay

An Amplex Red Cholesterol Assay Kit was purchased from Life Technologies (Invitrogen, Carlsbad, CA, USA). Manufacturer protocols were followed to prepare the standard curve and quantify cholesterol. Standard curves were prepared by preparing cholesterol concentrations of 0–20 μM (0–8 μg/mL). Viral particles were separated from the cholesterol-bound CD formulations via 50 kDa centrifugal filter units. Cells were pelleted by centrifugation at 2000 rpm for 5 min after CD treatment. Fluorescence was measured using a Biotek Synergy H1 microplate reader at an Ex/Em of 560/590. Background fluorescence determined for the cholesterol control reaction has been subtracted from each value.

### 4.8. Virus-Treated CD Studies

The rVSV-based SARS-CoV-2 pseudotype (Wuhan variant) (20,000 RLU) was incubated with 0.5–10% *w*/*v* formulations of either HPBCD or HPGCD for 2 h prior to the addition of the virus CD complex to Vero-E6 ACE2 cells (4 × 10^4^ cells/well in a 96-well plate). The pseudotyped virus was left to infect the cells for 8 hr prior to replacement with a complete cell culture medium. After 24 h of infection, cells were lysed using 1X cell culture lysis buffer (Promega), and 100 uL of luciferase substrate (Promega) was added based on the manufacturer’s instructions prior to luminescence readout using the Synergy H1 plate reader (Biotek, Winooski, VT, USA).

### 4.9. In Silico Molecular Docking

Substrates and receptors were modified using AutoDock Tools 1.5.6. The SARS-CoV-2 S protein from different variants (Wuhan (D614G), Delta, Epsilon (Cal.20), Gamma (P.1), Alpha (B.1.1.7), and Beta (B.1.351)) in their C1 binding state was obtained from Protein Data Bank (PDB). Binding simulations to HPBCD and HPGCD were conducted using AutoDock Vina (Scripps Research Institute, La Jolla, CA, USA). Two simulations were conducted for each ligand–variant combination due to the size of the protein and limitations within the maximum search space of the software. The S protein head comprised the S1, and part of the S2 subunit was categorized as the “top” portion of the S protein, whereas the S2 stalk comprised the “bottom” of the S protein in these simulations. Binding scores (binding free energy in kcal/mol) for the most favorable binding modes are recorded, as well as the relative binding affinity (ΔG) and dissociation constant (K_d_).

### 4.10. GFP Lentivirus Infection Studies

Studies assessing the efficacy of HPBCD against lentiviral infectivity were conducted using a pLV[Exp]-Puro-CMV > EGFP lentivirus (VectorBuilder, Chicago, IL, USA). HEK293T cells were seeded in 96-well plates at a density of 1 × 10^4^ cells/well. Cells from the untreated control wells were detached using 0.25% Trypsin-EDTA and counted prior to plating of the lentivirus to determine the volume of lentiviral addition for a multiplicity of infection (MOI) of 5. Polybrene was added at a final concentration of 5 μg/mL to enhance lentivirus uptake. The lentivirus was then added to the treated and non-cyclodextrin-treated HEK293T cells for 0.5–6 h and incubated at 37 °C and 5% CO_2_. After treatment, cells were washed twice with 1X DPBS, and complete cell culture media were added for an additional 24 h incubation. Fluorescence imaging was conducted using a BioTek Cytation 5 Cell Imaging Multi-Mode Reader. Fluorescence output was normalized to the untreated control and represented as percent relative fluorescence (RFU).

### 4.11. SARS-CoV-2 Culture

The Delta SARS-CoV-2 variant was obtained from the BEI repository (B.1.617.2, catalog: NR-55672). Vero E6 overexpressing ACE2 (Vero E6-hACE2) was obtained from Dr. Jae Jung and maintained in DMEM with high glucose, supplemented with 10% FBS, 2.5 μg/mL puromycin at 37 °C, and 5% CO_2_. SARS-CoV-2 virus from BEI was cultured and passaged four times in Vero E6-hACE2 cells and harvested every 48 h post-inoculation and tittered for plaque-forming units/mL of supernatant.

Plaque assays to quantify the production of live viruses were performed by seeding Vero-E6 ACE2 overexpressing cells (VeroE6-hACE2) in 6-well plates and incubating overnight at 37 °C. Culture media were removed the following day and replaced with fresh media, a 10-fold serial dilution of the culture media from the infected cells; cells were incubated for 45 min at 37 °C. Infectious material was removed, and a solution of DMEM and 2% agarose at a 1:1 ratio was added to each well. Following a 3-day incubation at 37 °C, the cells were then fixed with 4% paraformaldehyde. To image the plaques, the agar overlay was removed, and a solution of 0.2% (*w*/*v*) crystal violet was added to each well. The plate was then incubated for 5 min, crystal violet was removed, and the cells were washed with dH2O to remove residual excess crystal violet. Plaques were counted to determine PFU. Viral stocks were stored at −80 °C.

### 4.12. In Vitro/In Vivo Antiviral Activity Testing

Calu-3 cells were seeded in (2) 12-well plates at a density of 1 × 10^4^ cells/well and allowed to reach confluency prior to CD treatment and SARS-CoV-2 infection. Cells were pre-treated with either 2.5% HPBCD (*w*/*v*), 5% HPBCD, 10% HPBCD, 5% HPBCD + 0.5% CMC (50–200 cps), 5% HPBCD + 0.5% CMC (1500–2000 cps), or 5% HPBCD + 5% HPGCD for 1 h. PBS was used for the untreated control. Formulations were aspirated off and replaced with a complete cell culture medium. A viral titer for the SARS-CoV-2 (Delta variant) was quantified via a plaque formation assay and added to the cells at a multiplicity of infectivity (MOI) of 0.2 for 48 h. Cells were lysed with Trizol prior to removal from the BSL3 facility.

### 4.13. Gene Expression RT-PCR

Total RNA was extracted via Trizol (Invitrogen) following the manufacturer’s guidelines. Standard curves were prepared using the SARS-CoV-2 N Positive Control Plasmid (IDT). Viral target N protein was amplified (FAM) using the PrimeTime Gene Expression Master Mix (IDT). Viral load was quantified via qPCR using the QuantStudio 12K Flex system (Thermo Fisher, Waltham, MA, USA) based on the standard curve and reported as genomic copies/ug RNA or normalized to the untreated control.

Forward and reverse primers were designed using NCBI primer blast. Applied Biosystems (Waltham, MA, USA) PowerUp SYBR Green Master Mix was used for amplification of targets. Thermal parameters of RT-PCR amplification are as follows (standard cycling mode): 50 °C 2 min for UDG activation, and then 40 cycles of PCR (95 °C 2 min, 60 °C for 15 s). The reaction volume was 10 uL/well run on a 384-well plate. An RT-PCR assay was performed on a Thermo Fisher QuantStudio 6 Flex system. Sequences of primers are available upon request. Samples with Ct < 37 were assessed as positive, and samples > 40 were negative/undetectable.

### 4.14. In Vivo K18-hACE2 Study

Heterozygous K18-hACE2 mice (B6.Cg-Tg(K18-ACE2)2Prlmn/J) were obtained from The Jackson Laboratory. Animals were housed in groups and fed standard chow diets. All mice used in the study were male. Animal studies were carried out in accordance with recommendations and guidelines from the Guide for the Care and Use of Laboratory Animals of the National Institutes of Health. All protocols were approved by USC’s Institutional Animal Care and Use Committee (IACUC; ID# 21410). Viral inoculations were performed under anesthesia that was induced and maintained with ketamine hydrochloride. Eight-week-old male mice were intranasally pre-treated with 25 uL of HPBCD formulations/nostril once daily on day −1 and day 0 before intranasal inoculation with SARS-CoV-2 Delta variant at 10^4^ PFU. Weight and behavior were monitored over the course of this study. Mice were euthanized at 5 dpi. A timeline of the study conducted can be found in Appendix A. 

### 4.15. ACE2 ELISA

Protein was extracted from mouse samples via Trizol (Thermo Fisher Scientific, Waltham, MA, USA) and quantified using Bradford reagents (BioRad, Irvine, CA, USA); absorbance was read at 595 nm. The human ACE2 ELISA kit (ab285246, E4528) was purchased from Abcam (Cambridge, UK). Standards were prepared, and the assay was run according to the manufacturer’s instructions. Following the addition of Stop Solution, an endpoint reading was recorded at 450 nm.

### 4.16. Hematoxylin and Eosin Staining

The lung section slides were stained using a Hematoxylin and Eosin Stain Kit (Vector Labs, Newark, CA, USA, Catalogue No. H-3502). The staining was performed according to the manufacturer’s instructions, and the H&E-stained slide was observed (and photographed if necessary) as previously published [63].

### 4.17. Statistical Analysis

Statistical analyses were performed using Prism 8.0 software (GraphPad). Details on statistical analysis conducted for each figure are provided in the corresponding figure captions. The significance level of the mean differences was set at *p* < 0.05.

## 5. Conclusions

These in silico, in vitro, and in vivo studies provide evidence for the application of beta-cyclodextrin-based formulations to prevent viral infections. Here, we show that not all cyclodextrins exert the same effects across SARS-CoV-2 variants. Molecular docking studies identified that HPGCD has reduced binding affinity to spike protein variants compared to HPBCD. In addition to preventing SARS-CoV-2 infection, we have identified the potential of HPBCD as an agnostic barrier capable of inhibiting lentivirus, MHV, and H1N1 infections. Treatment with 5% HPBCD + 0.5% CMC (*w*/*v*) was able to reduce infection of the SARS-CoV-2 Delta variant by 94%, as quantified by qRTPCR in murine lung tissue. In summary, this report suggests the novel usage of HPBCD-based formulations for the prevention of viruses, such as SARS-CoV-2, through an intranasal approach. Additional in vivo studies affirm the safety and efficacy of HPBCD as an antiviral barrier when administered intranasally in the K18-hACE2 murine model. We propose that intranasal formulations based on HPBCD could provide additional benefits in preventing respiratory viral infections.

## Figures and Tables

**Figure 1 ijms-25-02061-f001:**
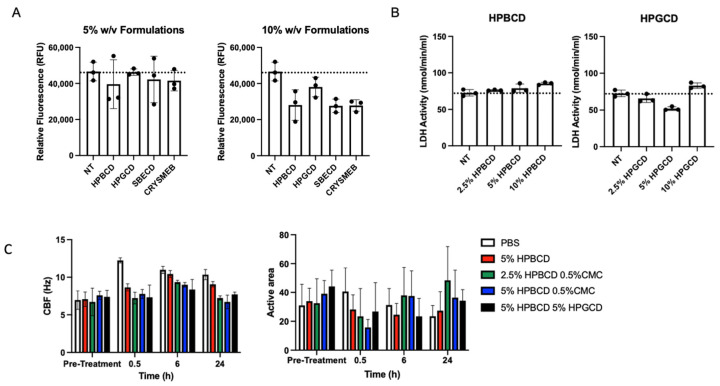
Candidate cyclodextrin safety profiles. In vitro cytotoxicity studies were conducted using HPBCD, HPGCD, SBECD, and CRYSMEB. (**A**) A549-ACE2 cells were treated with hydroxypropyl beta cyclodextrin (HPBCD), hydroxypropyl gamma cyclodextrin (HPGCD), sulfobutylether beta cyclodextrin (SBECD), or KLEPTOSE CRYSMEB, a low methylated beta cyclodextrin (MBCD), for 24 h. CellTox Green dye and an assay buffer were added to make a 2X 1:500 dilution. Fluorescence readout was measured (485–500 nm_Ex_/520–530 nm_Em_). Cytotoxicity data (n = 3) per CD treatment are shown as individual points on a bar graph. (**B**) A549-ACE2 cells were exposed to varying % *w*/*v* cyclodextrin (HPBCD, HPGCD) treatments for 48 h. (**C**) Ciliary beat frequency (Hz) pre- and post-treatment and percentage of the active area of ciliary beating. NT, no treatment; PBS, phosphate-buffered saline; CMC, carboxymethyl cellulose.

**Figure 2 ijms-25-02061-f002:**
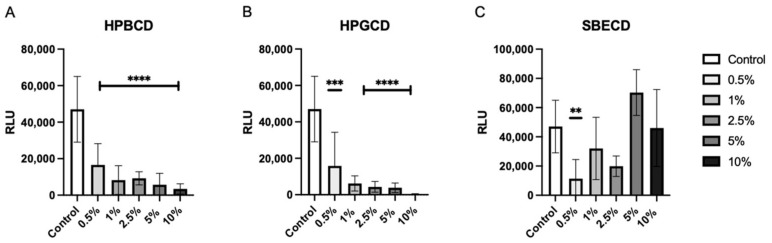
HPBCD and HPGCD inhibit the SARS-CoV-2 pseudotype in a concentration-dependent manner. A549-ACE2 cells were pre-treated with CD formulations (**A**) HPBCD, (**B**) HPGCD, or (**C**) SBECD for 1 h. Cells were washed 3X with PBS before replacement with a complete cell culture medium. Cells were infected with the SARS-CoV-2 Wuhan (D614G mutant) pseudotype for 8 h. Twenty-four hpi cells were lysed for luciferase readout. The control is the no-treatment comparator. ** *p* < 0.01, *** *p* < 0.001, **** *p* < 0.0001.

**Figure 3 ijms-25-02061-f003:**
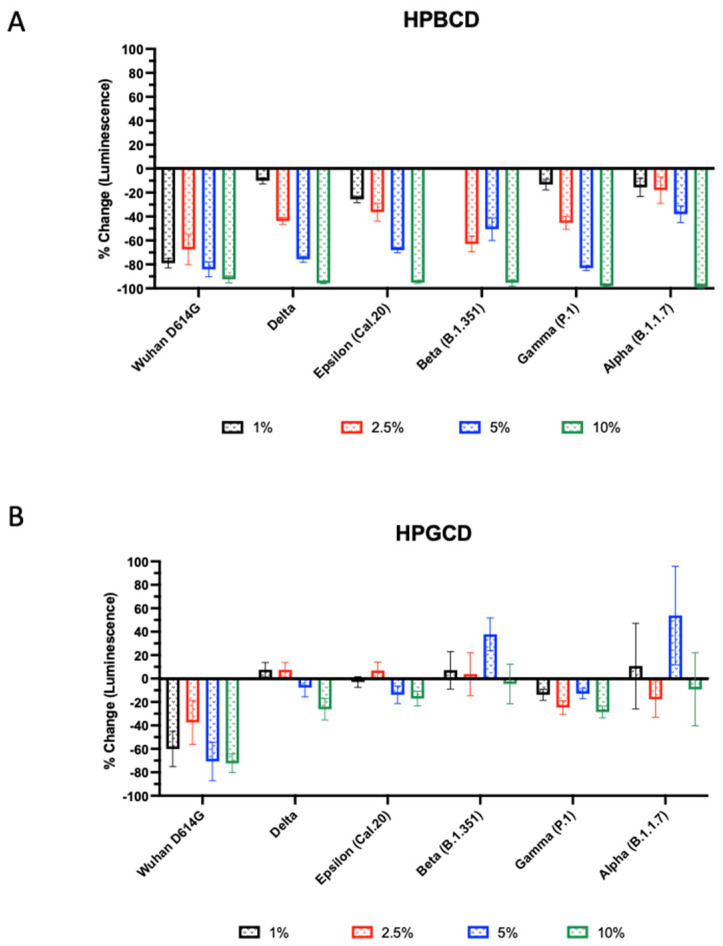
Beta- and gamma-cyclodextrin effect across SARS-CoV-2 pseudotyped variants. SARS-CoV-2 pseudotype infection of A549-ACE2 cells. Effect of (**A**) HPBCD and (**B**) HPGCD against Wuhan D614G, Delta, Epsilon, Beta, Gamma, and Alpha variants represented as percent change compared to the untreated control. Bars above zero indicate an increase in pseudotype infection for those variant/treatment groups.

**Figure 4 ijms-25-02061-f004:**
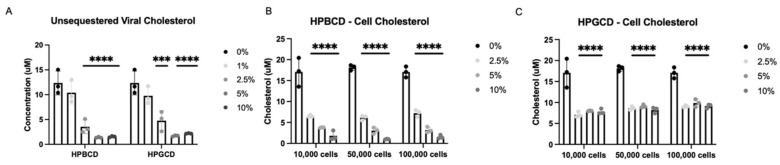
Cyclodextrin sequestration of viral and cellular cholesterol. (**A**) Unsequestered viral-bound cholesterol was quantified after 2 h of treatment with either HPBCD or HPGCD and separation of the virus CD complex via a 50 kDa centrifugal filter unit. Detection of cholesterol was quantified using the Amplex Red reagent-based assay. Fluorescence was measured with a microplate reader using an excitation/emission of 560/590. (**B**) Unsequestered cell membrane-bound cholesterol was quantified after 2 h of treatment with 0–10% *w*/*v* HPBCD. (**C**) Cell membrane-bound cholesterol was quantified after 2 h of treatment with 0–10% *w*/*v* HPGCD. One-way ANOVA followed by Dunnett’s test for multiple comparisons to a common control group revealed treatment-based statistical significance (*** *p* < 0.001, **** *p* < 0.0001).

**Figure 5 ijms-25-02061-f005:**
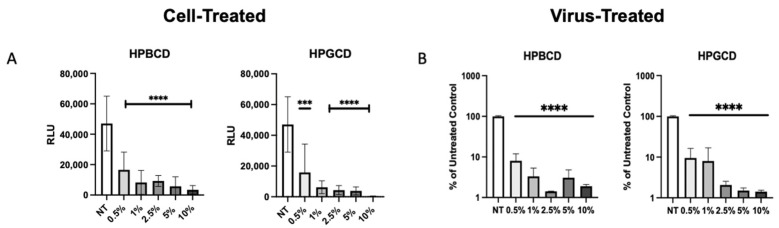
Cell-treated versus pseudotyped virus-treated effects of HPBCD and HPGCD. (**A**) Cyclodextrin effect against Wuhan SARS-CoV-2 pseudotype infection compared to the untreated control. rVSV-SARS-CoV-2-Luc (Wuhan variant) infection of A549-ACE2 cells post-treatment with 0–10% *w*/*v* formulations of HPBCD and HPGCD; one-way ANOVA *** *p* < 0.001, **** *p* < 0.0001. (**B**) CD treatment of SARS-CoV-2 pseudotyped virus reduces cell infectivity. A total of 20,000 RLU of the rVSV-SARS-CoV-2 Luc pseudotype (Wuhan variant) was incubated with 0.5–10% *w*/*v* treatments of HPBCD or HPGCD prior to infection of A549-ACE2 cells. Results plotted as a % of the untreated media control. Data are shown as mean (SD) (n = 3). Statistical analysis was conducted using one-way ANOVA (**** *p* < 0.0001), where the mean of each column was compared to the no-treatment (NT) group.

**Figure 6 ijms-25-02061-f006:**
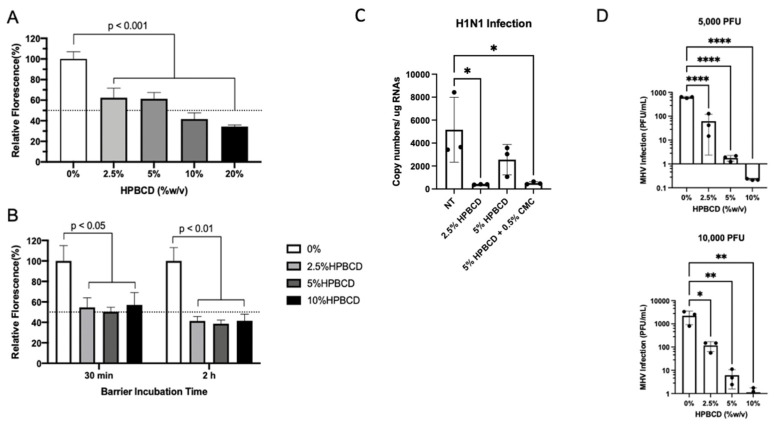
HPBCD is effective as an agnostic barrier against viral infections. HEK293T cells were infected with pLV[Exp]-Puro-CMV > EGFP for (**A**) 6 h in the presence of 0–20% HPBCD (*w*/*v*) and (**B**) 30 min or 2 h in the presence of 0–10% *w*/*v* HPBCD, both at an MOI of 5. Bars indicate the percent relative fluorescence, which reflects the GFP output from viral infection (n = 3). (**C**) WT A549 cells treated for 1 h prior to infection with H1N1 (A/PR/8/34) at an MOI = 0.2. Viral load quantified via qRTPCR 48 hpi. (**D**) Murine hepatitis virus (MHV) infection of NCTC clone 1469 in the presence of 0–10% *w*/*v* HPBCD treatment. Viral RNA levels were quantified based on a standard curve (0–1 × 10^6^ pfu/mL) prepared with known viral stocks. Statistical analyses: one-way ANOVA, * *p* < 0.05, ** *p* < 0.01, **** *p* < 0.0001.

**Figure 7 ijms-25-02061-f007:**
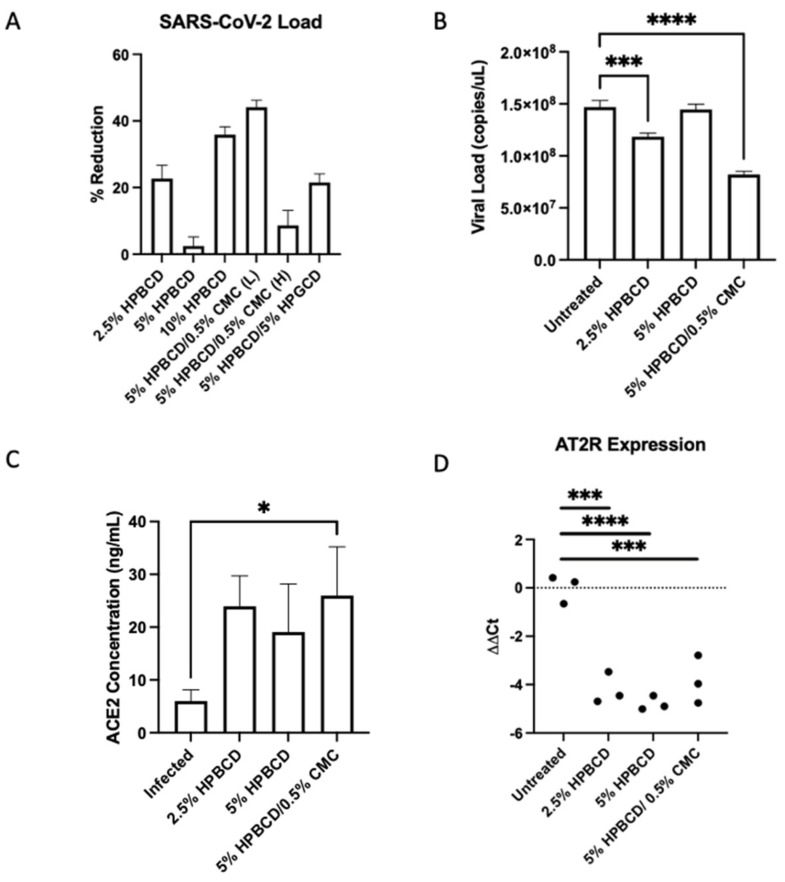
Cyclodextrin treatment reduces SARS-CoV-2 infection in human lung epithelial cells. (**A**–**D**) Calu-3 cells were seeded to confluency and pre-treated with CD formulations for 1 h prior to infection with SARS-CoV-2 (Delta B.1.617.2) at a multiplicity of infection (MOI) of 0.2. (**A**) Viral RNA levels were analyzed 48 h post-infection via qRTPCR. Bars indicate the mean with SD (n = 3). Data are presented as a percent reduction compared to infected positive control. (**B**) Viral N copies quantified via qRTPCR (n = 3 per group). (**C**) Cellular ACE2 protein concentrations were measured via ELISA; bars indicate the mean with SD. (**D**) AT2R ΔΔCt values across groups. Dotted horizontal lines indicate the baseline in SARS-CoV-2-infected cells without treatment; * *p* < 0.05, *** *p* < 0.001, **** *p* < 0.0001.

**Figure 8 ijms-25-02061-f008:**
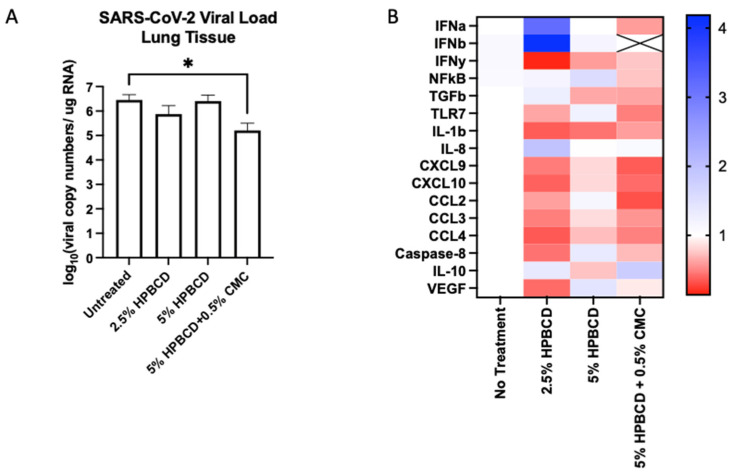
Cyclodextrin treatment effect on SARS-CoV-2 infection in K18-hACE2 mice. (**A**,**B**) Eight-week-old male K18-hACE2 transgenic mice were treated intranasally with CD-based treatments once daily for 2 days prior to intranasal viral inoculation (SARS-CoV-2 Delta variant) at 1 × 10^4^ PFU (n = 5 per group). Study design and weight monitoring are found in the Appendix A. (**A**) Viral load from murine lung tissue was quantified via qRT-PCR. (**B**) Fold change in the gene expression levels of inflammatory markers was normalized to beta-actin and compared to the no-treatment group. ** p* < 0.05.

**Figure 9 ijms-25-02061-f009:**
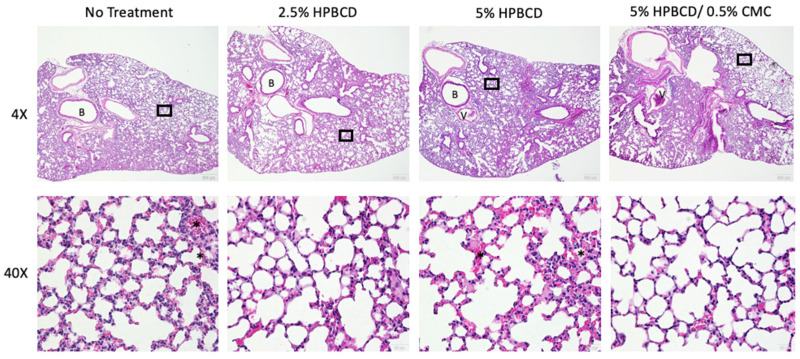
Histopathological alterations following SARS-CoV-2 infection. Hematoxylin and eosin staining of lung sections from K18-hACE2 mice following intranasal infection with 1 × 10^4^ PFU SARS-CoV-2 5 dpi; 4X low-resolution (top row) and 40X high-resolution (bottom row) representative images with 200 μm and 20 μm scale bars, respectively. Inflammatory cell accumulation and hemorrhage in alveolar spaces (asterisk) at 5 dpi. B: bronchiole; V: blood vessel. Black squares indicate regions of the lung tissue that were zoomed in for the corresponding 40X image.

**Table 1 ijms-25-02061-t001:** Characteristics of lead cyclodextrin compounds.

Cyclodextrins	Nominal Molecular Weight (g/mol)	Inner Cavity Diameter (A)	Inner Cavity Diameter (A)	Cavity Volume (A^3^)	Number of Subunits	R-Group(s)
HPBCD ^1^	1501	6.0–6.4	15.4	262	7	R = -H or [CH_2_CH(CH_3_)O]_n_H
SBECD ^2^	2242					R = -H or CH_2_CH_2_CH_2_CH_2_SO_3_Na
CRYSMEB ^3^	1191					R = -H or -CH_3_
HPGCD ^4^	1576	7.5–8.3	17.5	427	8	R = [CH_2_CH(CH_3_)O]_n_H, n = 0, 1, 2

^1^ HPBCD (2-hydroxypropyl beta-cyclodextrin); ^2^ SBECD (sulfo-butyl ether beta-cyclodextrin); ^3^ CRYSMEB (methyl beta-cyclodextrin); ^4^ HPGCD (2-hydroxypropyl gamma-cyclodextrin).

**Table 2 ijms-25-02061-t002:** Molecular docking of spike protein interactions with cyclodextrin. Docking results of HPBCD and HPGCD to the SARS-CoV-2 S protein S1/S2 subunit (“upper”) and S2 subunit (“lower”) search space regions across variants. PDB codes: 7LWT (Alpha), 7LYN (Beta), 7M8K (Gamma), 7TOV (Delta), 7N8H (Epsilon), 7KDL (Wuhan). The search space region is highlighted in the cube.

Search Space Region	S1/S2 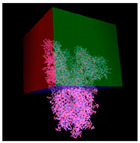			S2 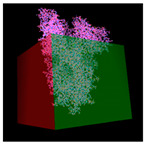		
Variant	Ligand	Binding Score (Kcal/mol)	K_d_ (μM)	Ligand	Binding Score (Kcal/mol)	K_d_ (μM)
Wuhan	HPβCD	−6.9	8.70	HPβCD	−7.4	3.74
	HPγCD	−7.1	6.21	HPγCD	−6.0	39.79
Alpha	HPβCD	−6.6	14.45	HPβCD	−6.9	8.70
	HPγCD	−6.6	14.45	HPγCD	−5.8	55.77
Beta	HPβCD	−6.8	10.31	HPβCD	−6.4	20.25
	HPγCD	−6.9	8.70	HPγCD	−6.4	20.25
Delta	HPβCD	−6.8	10.31	HPβCD	−7.6	2.67
	HPγCD	−6.8	10.31	HPγCD	−6.0	39.79
Epsilon	HPβCD	−8.1	1.15	HPβCD	−6.2	28.38
	HPγCD	−6.4	20.25	HPγCD	−6.5	17.10
Gamma	HPβCD	−6.6	14.45	HPβCD	−6.2	28.38
	HPγCD	−7.3	4.43	HPγCD	−6.8	10.31

## Data Availability

The data that support the findings of this study are available from the corresponding author upon reasonable request.

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
