# Peer review of "Hydroxypropyl-Beta Cyclodextrin Barrier Prevents Respiratory Viral Infections: A Preclinical Study"

_ijms, 2024, doi:10.3390/ijms25042061_

Round 1
Reviewer 1 Report
Comments and Suggestions for Authors
Dear Authors,
Congratulations on your valuable and interesting work!
Please answer to my questions:
1. How did you prepare the CD solutions? Have you measured the pH and osmolarity? Maybe it has important role because in the nose the preferred pH around 7, the movement of cilia is affected also by pH.
2. At some sencences additional spaces can be seen, e.g. line 41, 71, 249, 256, 319, 333, 464, 596
3.SBECD is an ionic CD, because the sulfonate group and its sodium salt is linked to the cyclodextrin cavity by a butyl chain. Therefore it is different from the others. I think this information should be described in the papaer. Do you think it has an effect on the mechanism of action on antiviral activity?
4. Figure 1. Indicate NT meaning. How do you explain that the 10% CD sample has higher LD activity? You said: "Com- 261 pared to the no-treatment control, HPBCD and HPGCD did not increase lactate dehydro- genase activity, indicative of cytotoxicity."
5. Table 2. should start on page 11.
6. Label of Figure 5 and 7: some letters/raws are smaller, some is bold. Please correct it. and Figure 8. the first sencence is bold. Figure 9. is bigger font size.
7. Figure 8: Why CMC formulation is more effectice than CD alone? Please explain it in more details.
8. You did not mention SBECD in the Discussion and Conclusion, please explain why did you stop investigate this CD.
Author Response
Our responses to your comments are attached.

Reviewer 2 Report
Comments and Suggestions for Authors
It has been reported by other studies (PMID: 37311279, PMID: 37242317, PMID: 36080368, PMID: 35798224) on the antiviral effects of (Hydroxypropyl)-beta cyclodextrin on SARS-CoV-2, which dramatically compromises the novelty of this manuscript though a small part of anti H1N1 and MHV studies were included in this manuscript. It will be more interesting if detailed mechanisms by which Hydroxypropyl-beta cyclodextrin exhibits antiviral effects against different viruses are investigated. It will also have more significance if a comprehensive evaluation of the antiviral functions of Hydroxypropyl-beta cyclodextrin in animal models.
In this manuscript, there are some mistakes or missing contents, for example:
1- DNA copies/ul, lobe (Fig 6C and Fig.8A)?
2- There is no information about live SARS-CoV-2 and how to titrate
…
Author Response

(The authors gave the same response as above.)

Reviewer 3 Report
Comments and Suggestions for Authors
The presented data strengthens the relevance of novel strategies to fight SARS-CoV-2, which might include cyclodextrins, including various derivatives like hydroxypropyl beta-cyclodextrin (HPBCD) and hydroxypropyl gamma-cyclodextrin (HPGCD). While cyclodextrins, including HPBCD and HPGCD have been widely studied for various applications, there are some potential disadvantages and considerations regarding the effects of cyclodextrins on biological functions of different cells.
Overall, this manuscript is well-written. I have minor comments that need to be addressed:
1. Please define any symbols, abbreviations in different figures/write Figure Legends more effectively.
2. Please highlight limitation and considerations associated with the use of cyclodextrins (e.g., solubility, stability, cytotoxicity, complexation efficiency, influence on signaling pathways, cholesterol depletion, in vivo pharmacokinetics, etc.) and explain the gaps your study filled in previous scientific knowledge and what are the implication for researcher in the same field for the future studies.
Comments on the Quality of English LanguageThe quality of English language is fine.
Author Response
Our comments are attached.

Reviewer 4 Report
Comments and Suggestions for Authors
The authors have executed excellent work on developing hydroxypropyl-beta cyclodextrin and CMC containing formulations with distinct concentrations and compositions. The beta cyclodextrin containing formulations demonstrated superior anti-viral properties compared to other formulations. The manuscript is well written, and needs minor inclusion of the following request:
1. Since the hydrogel formulation containing CMC will erode over a period, what will the frequency of administration of these formulations if advanced to clinical trials?
2. Please include the prophylactic potential of these formulations for preventing new viral infections in the manuscript. Please include the dose adjustment strategy for regions of higher infection and probability of inter-personal variability while designing the right dose of the optimum formulation.
Author Response
Our response to your comments are attached.

Round 2
Reviewer 2 Report
Comments and Suggestions for Authors
Reviewer # 2
It has been reported by other studies (PMID: 37311279, PMID: 37242317,
PMID: 36080368, PMID: 35798224) on the antiviral effects of
(Hydroxypropyl)-beta cyclodextrin on SARS-CoV-2, which dramatically
compromises the novelty of this manuscript though a small part of anti
H1N1 and MHV studies were included in this manuscript. It will be more
interesting if detailed mechanisms by which Hydroxypropyl-beta
cyclodextrin exhibits antiviral effects against different viruses are
investigated. It will also have more significance if a comprehensive
evaluation of the antiviral functions of Hydroxypropyl-beta cyclodextrin in
animal models.
Please address my concern regarding the novelty of your study carefully. Compared to very related previous publications, what is the new information/significance in your study?
In this manuscript, there are some mistakes or missing contexts, for
example:
1 - DNA copies/ul, lobe (Fig 6C and Fig.8A)?
Legend errors have been corrected. ( I do not think so, please double check Fig.8A. SARS-CoV-2 is RNA virus)
2- There is no information about live SARS-CoV-2 and how to titrate
A new section 2.11. has been added on SARS-CoV-2 viral titration in the methods section.
(the information about live SARS-CoV-2 (which strain, how to propagate the virus) is still missing.
Author Response
Please address my concern regarding the novelty of your study carefully. Compared to very related previous publications, what is the new information/significance in your study?
While other groups such as Bezerra et al. (2022) have published on HPBCD inhibition of SARS-CoV-2 replication and virus-induced inflammatory cytokines and Alboni et al. (2023) have published on HPBCD depletion of membrane cholesterol and inhibition of SARS-CoV-2 entry into HEK293T-ACE2 cells, our manuscript is distinct in that it looks at:
- an in vitro characterization of different CDs which shows that not all cyclodextrins exert the same preventative effects;
- the optimal concentrations of CDs that will combine anti-viral efficacy with safety to guide the translation into the clinic and
- an in vivo assessment of HPBCD preventing SARS-CoV-2 Delta variant infection through intranasal administration.
Prior art and publications have relied heavily on the formulation capabilities of CDs and the blanket claim of anti-viral activity of CDs. However, the novelty in our approach lies in the application of HPBCD as a barrier coating on susceptible regions of the nasal cavity to prevent respiratory viral infection. Our data presented in this manuscript shows the safe and effective concentration of HPBCD with(out) a thickening agent. We have also shown efficacy in an animal model paving way for our team to advance into clinical testing quickly. This data supplements our patent (US Patent App. 17/996,452) and another publication by our team (Du, J., Shao, X., Bouteiller, JM.C. et al. Computational optimization of delivery parameters to guide the development of targeted Nasal spray. Sci Rep 13, 4099 (2023). https://doi.org/10.1038/s41598-023-30252-4) which utilizes in silico modeling to optimize a nasal spray delivery system to translate the discovery.
In this manuscript, there are some mistakes or missing contexts, for
example: 1 - DNA copies/ul, lobe (Fig 6C and Fig.8A)?
Legend errors have been corrected. ( I do not think so, please double check Fig.8A. SARS-CoV-2 is RNA virus)
We acknowledge that SARS-CoV-2 is an RNA virus. However, to quantify viral load, we converted RNA to cDNA and used a DNA positive control: 2019-nCoV_N_Positive Control (IDT, catalog #10006625), which consists of a plasmid containing the complete N gene (1,260 base pairs) of SARS-CoV-2. As such, the legend was labeled as DNA copies.
We have changed the y-axis further to “copy numbers/ug RNA” to clarify and prevent any further confusion regarding units. This was done for both Figure 8A and for Figure 6C. We hope this clarifies any concerns regarding the quantification labeling.
2 - There is no information about live SARS-CoV-2 and how to titrate
A new section 2.11. has been added on SARS-CoV-2 viral titration in the
methods section. (the information about live SARS-CoV-2 (which strain, how to propagate the virus) is still missing.
The following has been added to the manuscript methods under section 2.11 line 193:
“The Delta SARS-CoV-2 variant was obtained from the BEI repository (B.1.617.2, catalog: NR-55672). Vero E6 overexpressing ACE2 (Vero E6-hACE2) were obtained from Dr. Jae Jung and maintained in DMEM high glucose, supplemented with 10% FBS, 2.5 ug/mL puromycin at 37°C, 5% CO2. SARS-CoV-2 virus from BEI was cultured and passaged four times in Vero E6-hACE2 cells and harvested every 48 hours post-inoculation and tittered for plaque-forming units/mL of supernatant.”
Round 3
Reviewer 2 Report
Comments and Suggestions for Authors
Thank you for the explanation, but it is not a proper way to use DNA copies in Figs when showing RNA virus genome copies. Besides, ”Viral concentrations” (line 230) is not accurate.
Please check the paper (PMID: 32972994) and revise the labels (in the Figs) and the text in the manuscript.
There are several publications on the antiviral effects of (Hydroxypropyl)-beta cyclodextrin on SARS-CoV-2, which are very related to the current study. In the introduction part, none of them was mentioned. In the discussion part, only one publication has been cited (lines 612 to 614. In the last comments (Please address my concern regarding the novelty of your study carefully. Compared to very related previous publications, what is the new information/significance in your study?), My suggestion is that the authors should address this novelty concern properly in the manuscript ( for example, in the discussion part or introduction part )and very clearly show readers what the new information is in your study since several related publications have been published.
Line 46, “Although SARS-CoV-2 has primarily been linked to severe downstream respiratory,” Is inaccurate. It depends on what kinds of SARS-CoV-2.
Line 53. ” ..reveal ACE2 and its associated transmembrane protease serine 2 (TMPRSS2) to…” Does ACE2 associate with TMPRSS2?
Author Response
Attached is our response. Thank you.
